# Transcriptome Analysis of Ethylene Response in *Chrysanthemum moriflolium* Ramat. with an Emphasis on Flowering Delay

Hua Cheng, Min Zhou, Yuyang Si, Wenjie Li, Likai Wang, Sumei Chen, Fadi Chen and Jiafu Jiang *

State Key Laboratory of Crop Genetics & Germplasm Enhancement and Utilization, Key Laboratory of Landscaping, Ministry of Agriculture and Rural Affairs, Key Laboratory of Biology of Ornamental Plants in East China, National Forestry and Grassland Administration, Zhongshan Biological Breeding Laboratory, College of Horticulture, Nanjing Agricultural University, Nanjing 210095, China; chenghua0802@sina.com (H.C.); zm5206612@sina.com (M.Z.); thisyy@outlook.com (Y.S.); 2020204048@stu.njau.edu.cn (W.L.); wlk@njau.edu.cn (L.W.); chensm@njau.edu.cn (S.C.); chenfd@njau.edu.cn (F.C.)
* Correspondence: jiangjiafu@njau.edu.cn

**Abstract:** Ethylene is a gaseous phytohormone that delays flowering in *Chrysanthemum morifolium* Ramat. To date, however, there have been no systematic studies on genes involved in the ethylene response of this species, and the mechanism underlying ethylene-delayed flowering remains unclear. Herein, we applied RNA sequencing to characterize the ethylene response by comparing the transcriptomes of chrysanthemum cultivar 'Jinba' with or without ethephon treatment. Six unique RNA-seq libraries were generated. The identified differentially expressed genes (DEGs) primarily involved ethylene, auxin, and abscisic acid signaling genes; circadian clock genes; genes encoding functional proteins associated with floral transition, such as homologs of *AP1/FRUITFUL-like 1* (*AFL1*), *TERMINAL FLOWER 1* (*TFL1*), and so on; and genes encoding transcription factors, specifically of the MYB and bHLH families. Furthermore, quantitative RT-PCR was used to verify the DEGs identified by RNA-seq. Heterologous *CmAFL1* overexpression in *Arabidopsis thaliana* resulted in early flowering. Our findings present a landscape of transcriptomes and reveal the candidate genes involved in the ethylene-mediated regulation of flowering time in chrysanthemum, providing useful data for further studies.

**Keywords:** ethylene; *Chrysanthemum*; transcriptome; floral induction; *CmAFL1*





## 1. Introduction

Floral transition is a significant developmental process in the life cycle of flowering plants [1]. Plants perceive diverse environmental and endogenous signals to ensure timely transition from vegetative growth to flowering [2,3]. Accordingly, plants have evolved intricate regulatory mechanisms of floral induction [4]. Over the past decades, an increasing number of studies have revealed the mechanisms underlying floral initiation in plants, particularly in the model species *Arabidopsis thaliana* [5]. To date, six genetic pathways regulating flowering in *A. thaliana* have been identified: photoperiod [6], autonomous [7], vernalization [8], gibberellin (GA) [9], temperature [10], and age [11] pathways. All of these pathways converge on several floral integrator genes, including *FLOWERING LOCUS T* (*FT*), *SUPPRESSOR OF OVEREXPRESSION OF CO1* (*SOC1*), and *AGAMOUS-LIKE 24* (*AGL24*), which activate the floral meristem identity genes *LEAFY* (*LFY*), *APETALA1* (*AP1*), *SEPALLATA3* (*SEP3*), and *FRUITFULL* (*FUL*), ultimately accomplishing the transition from vegetative to reproductive growth [12].

Ethylene is a multifunctional phytohormone involved in the regulation of stress response and development, including floral transition [13]. However, ethylene produces diverse impacts on the floral induction of plants, which vary across species. For instance,

ethylene promotes flowering in pineapple (*Ananas comosus* L.) [14]. Meanwhile, in *Arabidopsis thaliana* and rice (*Oryza sativa* L.), ethylene either promotes or inhibits flowering under different conditions [15–17]. A previous study showed that the activated CTR1/EIN3 module suppressed the levels of bioactive GA, promoted the accumulation of DELLAs, and repressed the expression of the floral meristem identity genes *LFY* and *SOC1*, ultimately delaying flowering [18]. Another recent study showed that ACC (a precursor of ethylene synthesis) treatment downregulated the transcription and protein accumulation of the histone demethylase *FLOWERING LOCUS D* (*FLD*), thereby enhancing the levels of H3K4me2 at the *FLC* site and activating the expression of *FLC* and its homologous genes to delay flowering [19]. Ethylene has been reported to suppress flowering in chrysanthemum (*Chrysanthemum morifolium* Ramat.) [20], which are some of the most important cut flowers in ornamental plants worldwide and economically significant species in the floriculture industry. However, the molecular mechanisms underlying ethylene-mediated flowering delay in chrysanthemums remain unclear.

Here, we explored the effects of exogenous ethephon (an ethylene-generating agent) treatment in the short-day (SD) chrysanthemum cultivar 'Jinba'. After confirming the late-flowering phenotype, we used RNA sequencing (RNA-seq) to acquire the transcriptomes of 'Jinba' plants with and without ethephon treatment before floral induction. The identified differentially expressed genes (DEGs) included phytohormone signaling-related genes, circadian clock genes, flowering genes, and genes encoding MYB and bHLH transcription factors (TFs). Moreover, we selected the candidate flowering gene *CmAFL1* for heterologous transformation into *A. thaliana*, which resulted in early flowering. Overall, this work builds a foundation for further studies on molecular mechanisms underlying ethylene-mediated inhibition of chrysanthemum flowering.

## 2. Materials and Methods

### 2.1. Plant Material and Grown Conditions

Cuttings of the *Chrysanthemum morifolium* cv. 'Jinba' were obtained from the Chrysanthemum Germplasm Resource Preserving Center (Nanjing Agricultural University, China). After rooting, the plants were transferred to plastic pots (diameter = 18 cm) containing a mixture of turf and organic fertilizer (1:3) and grown in a greenhouse under natural light plus a 2 h night break from 22:30 to 00:30 h with yellow fluorescent light (>70 $\mu mol \cdot m^{-2} \cdot s^{-1}$). When the plants grew 20 fully expanded leaves, they were transferred to a growth chamber (25 °C) and grown under natural short-day (SD) conditions for water and ethephon treatment.

The *A. thaliana Columbia* ecotype (*Col*) was used as the wild type. The background of all overexpression lines was derived from *Col*. The plants were maintained in a growth chamber at 22 °C under long-day (LD) conditions (16 h light/ 8 h dark).

### 2.2. Ethephon Treatment

For ethephon treatment under SD conditions, 100 $mg \cdot L^{-1}$ ethephon was externally applied in the form of a spray to seedlings with up to 20 leaves every other day; all seedlings were treated five times. As controls, an equal number of seedlings were simultaneously sprayed with water (*n* = 57). The samples were harvested at 6 days after treatments. Each sample comprised three apical meristems and three of the third expanded leaf from the bottom of the apical meristem of the seedlings. Three biological replicates were set per treatment. The harvested samples were immediately wrapped in tinfoil, snap frozen in liquid nitrogen, and stored at −80 °C. A total of six samples were collected. Controls (labeled C) and ethephon-treated (labeled T) samples were designated C-1, C-2, and C-3 and T-1, T-2, and T-3, respectively. Six frozen samples were sent to Novogene Bioinformatics Technology Co., Ltd. (Tianjin, China) for RNA extraction and sequencing.

### 2.3. RNA Extraction and Transcriptome Sequencing

Total RNA from the apical meristems and leaves was extracted separately using a total RNA isolation system (TIANGEN, Beijing, China) according to the manufacturer's protocol. After confirming the RNA quality, cDNA libraries were constructed and sequenced using an Illumina platform. Briefly, 1 μg of total RNA from the apical meristems and leaves in each biological replicate of the samples was pooled at equal amounts and used as a single sample for transcriptome sequencing. The cDNA libraries were constructed from RNA samples for paired-end (PE) sequencing using the HiSeq™ Xten device (Illumina, San Diego, CA, USA) following the manufacturer's protocol.

### 2.4. De Novo Assembly, Gene Functional Annotation, and Differential Gene Expression Analysis

Raw reads were edited to remove the adaptors, reads with a greater proportion of unknown bases (N), and low-quality reads (>50% bases with small Qphred $\leq$ 5). De novo transcriptome assembly was performed using Trinity (v2.4.0) [21]. The assembled unigenes were analyzed using fragments per kilobase of transcript per million fragments mapped (FPKM) to determine their expression levels. Based on the results of DESeq [22], genes with the false discovery rate (FDR) below 0.05 and fold change in expression of 1.5 or higher were selected as DEGs. Functional annotation was performed based on homology search against the Nr, Nt, Pfam (protein family), KOG/COG, Swiss-Prot, KO, and GO databases using the BLASTx algorithm [23]. Principal component analysis (PCA) and KEGG enrichment analysis were performed on BMK Cloud (www.biocloud.net, accessed on 17 February 2022); the heat maps and the volcanic map were generated using TBtools.

### 2.5. Quantitative RT-PCR (qRT-PCR)-Based Validation of DEGs

Four DEGs were selected for qRT-PCR validation. The primers (Table S1) were designed using Premier 5.0. *CmEF1α* (KF305681.1) was used as the reference gene. For each sample, the reaction mixture for qRT-PCR contained 10 μL of 2 × SYBR Green Master Mix Reagent (Takara, Beijing, China), 1 μL of gene-specific primers, and 25 ng of the cDNA template. The amplification protocol was as follows: 95 °C for 2 min, followed by 45 cycles of 95 °C for 10 s, 60 °C for 10 s, and 72 °C for 20 s. The expression levels of DEGs were calculated using the $2^{-\Delta\Delta Ct}$ method [24]. qRT-PCR was performed using three biological and three technical replicates for each treatment.

### 2.6. Floral Dip Transformation of Arabidopsis thaliana

We amplified the '35S + AtADH 5'UTR' sequence from the pDEST_35AA_SRDX_BCKH vector [25] with the primers SRDX-F/R (Table S1) and inserted it at the multicloning site *Sac* II and *Nhe* I of pORE-R4 (GenBank: AY562547.1) to obtain a 35S promoter-driven overexpression vector that we renamed as pORE-R4-35AA. We used the primer pair pORE-R4-35AA-*CmAFL1*-F/R (Table S1) containing the *Bam*H I and *Eco*R I restriction site to amplify the open reading frame (ORF) of *CmAFL1*. Then the plasmid of pORE-R4-35AA vector was digested by *Bam*H I and *Eco*R I to obtain the linearized vector fragment. Afterwards, the amplicons of *CmAFL1* and the linearized vector were recombined together via Recombinant enzyme Exnase II (Vazyme, Nanjing, China) to obtain pORE-R4-35AA-*CmAFL1* construct (35S::*CmAFL1*). The overexpression construct 35S::*CmAFL1* was transformed into *EHA105* for heterologous transformation of *A. thaliana* via floral dip [26]. Transgenic progenies were selected on solidified 1/2 MS medium containing 35 mg·$L^{-1}$ kanamycin. Thereafter, DNA was extracted using a rapid plant genomic DNA isolation kit (Sangon Biotech, Shanghai, China) according to the manufacturer's instructions. $T_1$ seedlings were identified at the DNA level using the primers *CmAFL1*-F and GFP-R (Table S1). RNA was extracted from $T_3$ homozygous plants and reverse-transcribed into cDNA for semi-quantitative analysis according to the manufacturer's instructions.

### 2.7. Data Analysis

Data were analyzed using SPSS 25 (IBM, New York, NY, USA). Significant differences were determined using Student's *t*-test ($p < 0.05$). Data are presented as mean ± standard deviation.

## 3. Results

### 3.1. Ethylene Suppressed Flowering in Chrysanthemum 'Jinba'

Ethylene plays pivotal roles in regulating the growth and development, including the flowering process, in both dicotyledonous and monocotyledonous plants [16–18]. In the present study, we sprayed chrysanthemum 'Jinba' seedlings with water or ethephon and noted that the seedlings treated with water exhibited visible flower buds at 79 days after plantation (Figure 1a), whereas the leaves of apical meristem were still unfolding in seedlings treated with ethephon (Figure 1b). After 99 days of plantation, the seedlings treated with ethephon started budding (Figure 1c). Collectively, these results indicate that ethylene inhibits chrysanthemum flowering, consistent with previous reports [27].

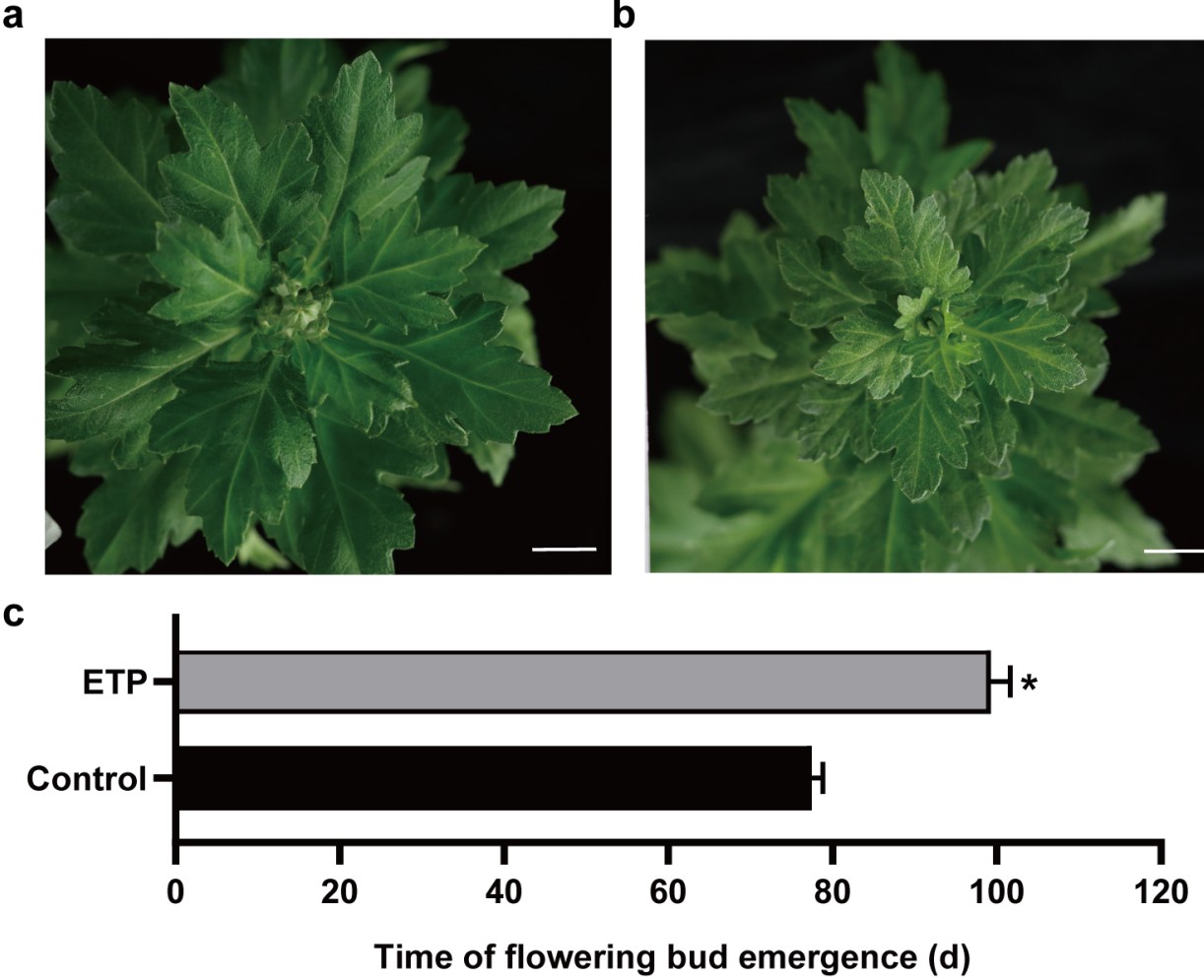

**Figure 1.** Phenotypes of wild type chrysanthemum 'Jinba' following water and ethephon treatment. Phenotypes of wild type chrysanthemum 'Jinba' treated with water (**a**) and ethephon (**b**) at 79 days after plantation. Bar = 1 cm. (**c**) Statistics of the time of flowering bud emergence. Student's *t*-test was used to verify significant differences (* $p < 0.05$); $n = 57$. Error bars indicate standard deviation (SD).

### 3.2. RNA-Seq after Ethylene Treatment

Based on the observed phenotypes, we performed RNA-seq to identify the key genes involved in ethylene-mediated inhibition of flowering. To identify the genes of interest, six cDNA libraries were constructed using RNA samples obtained from three water-treated (C-1, C-2, and C-3) and three ethephon-treated (T-1, T-2, and T-3) samples. The total amount of clean data obtained from RNA-seq of the six samples was 50.45 GB, with at least 7.20 GB of clean data obtained per sample. The mean GC content was 42.1%; furthermore, in each sample, over 91.66% of bases scored Q30 or higher (Table S2). The clean data were then mapped using HISAT2, with the mapping ratio ranging from 73.56% to 78.67% (Table S3). The Pearson's correlation coefficient of the three control samples ranged from 0.90 to 0.96, while that for the three ethephon-treated samples ranged from 0.86 to 0.94 (Figure 2a). PCA reduces the dimensionality of large datasets and improves interpretability, and it has been widely applied to visually evaluate RNA-seq results [28,29]. Therefore, we used PCA. The six control (C) and ethephon-treated (T) samples formed two groups, and samples of the same treatment were clustered together (Figure 2b), indicating excellent reproducibility of RNA-seq results between the biological replicates. Therefore, ethephon treatment was the key driving factor separating the datasets.

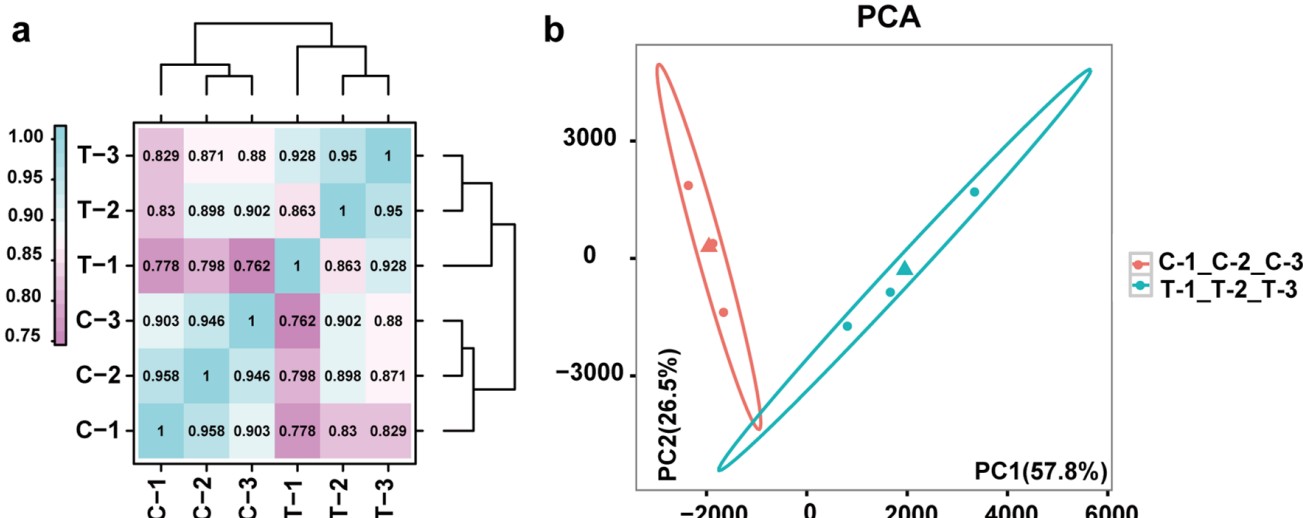

**Figure 2.** General analysis of the transcriptome data of samples following water and ethephon treatments. Pearson's correlation analysis (**a**) and principal component analysis (PCA) (**b**) between six samples. x- and y-axes represent each sample. C-1, C-2, and C-3 indicate three control samples. T-1, T-2, and T-3 represent three ethephon-treated samples.

### 3.3. Differential Gene Expression Analysis

DEGs were analyzed after standardizing the read count data using DESeq2 [30]. The screening criteria of DEGs analyzed in the present study were FDR < 0.05 and fold change ≥ 1.5. A total of 2630 genes were differentially expressed pairwise between the control and ethephon-treated samples; of these, respectively, 1363 and 1267 genes were up- and downregulated (Figure S1, Table S4).

To explore the functional categories and key biological pathways related to these DEGs, we performed KEGG enrichment analysis by selecting the top 30 KEGG terms with the lowest Q values. The genes were enriched in plant hormone signal transduction and circadian rhythm pathways (Figure 3), indicating that these pathways may be involved in ethephon treatment.

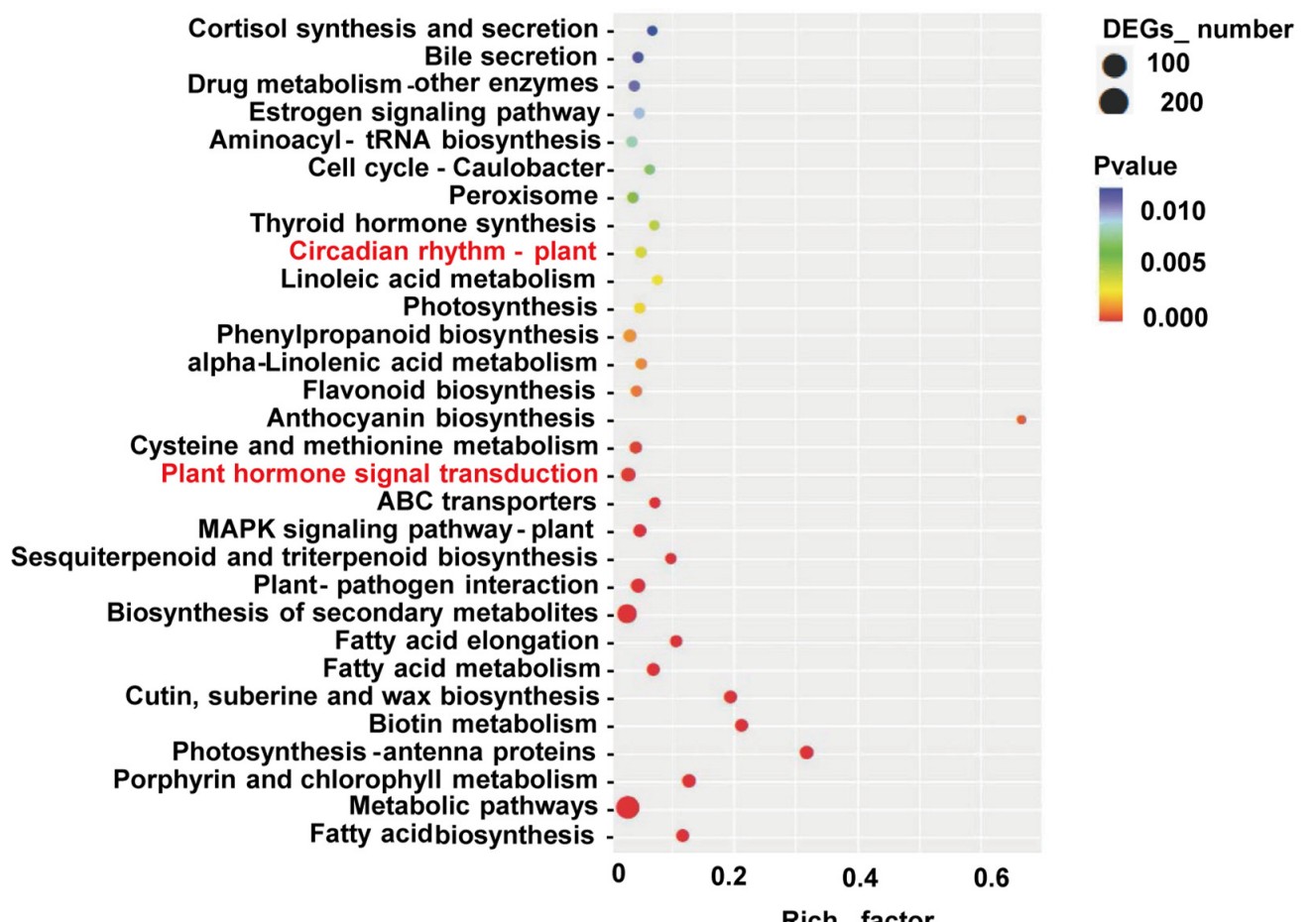

**Figure 3.** Analysis of differentially expressed genes (DEGs) between control (water) and ethephon-treated samples. KEGG enrichment analysis of 2630 DEGs. The x-axis indicates the enrichment ratio (enrichment ratio = term candidate gene number/term gene number); the y-axis represents the enriched KEGG pathways.

*3.4. DEGs Related to Phytohormone Signaling and Circadian Rhythm*

As 'plant hormone signal transduction' was identified as an enriched KEGG pathway, we analyzed DEGs related to phytohormone signaling. We identified certain genes involved in the ethylene signaling pathway: *ETHYLENE RESPONSE 2* (*ETR2*), *CONSTITUTIVE TRIPLE RESPONSE 1* (*CTR1*), and *EIN3-BINDING F-BOX PROTEIN 2* (*EBF2*) genes were upregulated by ethephon treatment, while most of the *ETHYLENE RESPONSE FACTOR* (*ERF*) genes were downregulated (Figure 4a). In addition, we identified many DEGs involved in the auxin signaling pathway, such as *AUX/INDOLE-3-ACETIC ACID 1* (*IAA1*) and *IAA6*, both of which were downregulated following ethephon treatment (Figure 4b). Moreover, the ABA receptor PYL2 and its downstream phosphatase protein PP2C involved in ABA signaling were also downregulated following ethephon treatment (Figure 4c). Overall, these results indicate that the auxin and ABA signaling pathways may be affected by ethylene to affect floral induction.

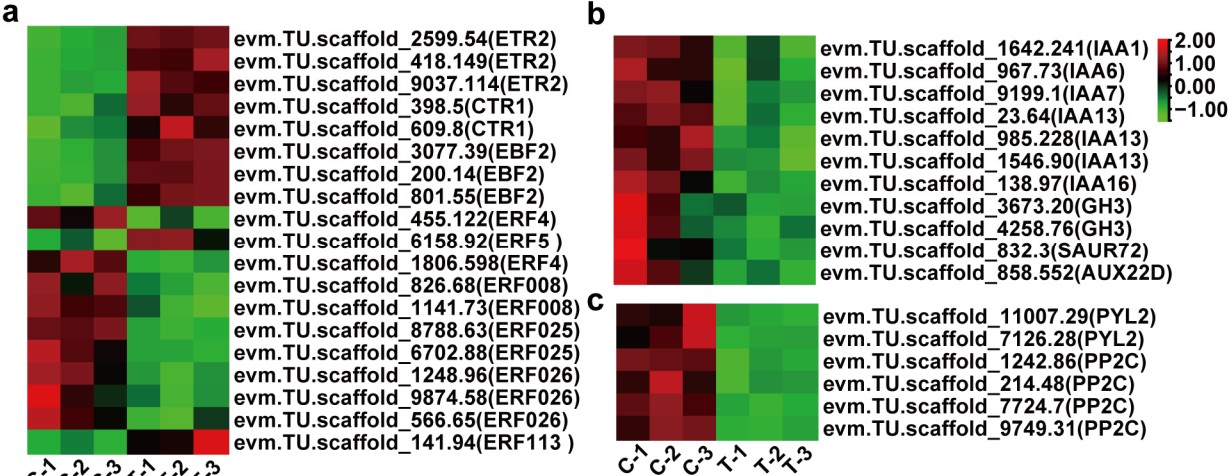

**Figure 4.** Heat map of differentially expressed genes (DEGs) related to ethylene (**a**), auxin (**b**), and ABA (**c**) signaling pathways between control (water) and ethephon treatments. C-1, C-2, and C-3 represent three control samples. T-1, T-2, and T-3 represent three ethephon-treated samples.

The circadian clock controls several important processes in plant development, including the transition from vegetative growth to floral transition [31]. Most of the circadian-regulated genes are components of the photoperiodic pathway. Here, we noted the circadian-rhythm-related genes *PHYTOCHROME A (PHYA)*, *A. thaliana PSEUDO-RESPONSE REGULATORS 7 (APRR7)*, E3 ligase *CONSTITUTIVE PHOTOMORPHOGENIC1 (COP1)*, and *GIGANTEA (GI)* were upregulated following ethephon treatment (Figure 5a). Therefore, ethylene may affect the circadian clock to regulate chrysanthemum flowering.

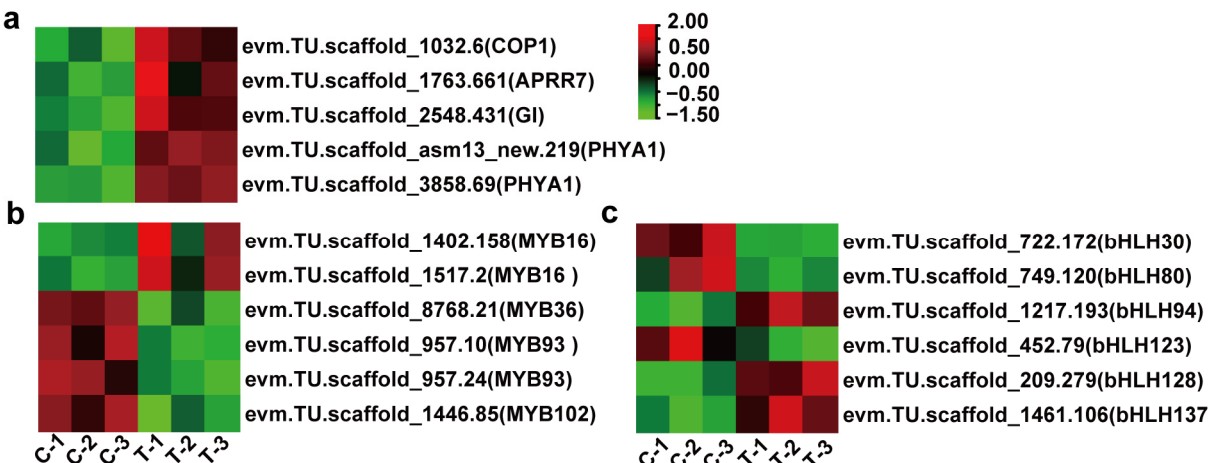

**Figure 5.** Heat map of differentially expressed genes (DEGs) related to the circadian clock (**a**), MYB TFs (**b**), and bHLH TFs (**c**) between control (water) and ethephon-treated samples. C-1, C-2, and C-represent three control samples. T-1, T-2, and T-3 represent three ethephon-treated samples.

*3.5. Floral Induction Genes and TFs Underlying in the Effect of Ethylene*

Since ethylene treatment inhibited chrysanthemum flowering, we focused on the differential expression of flowering genes. As expected, the chrysanthemum flowering integrator *AP1/FRUITFUL-like (AFL1)*, *SQUAMOSA PROMOTER-BINDING PROTEIN-LIKE 1 (SBP1)*, and *SQUAMOSA PROMOTER-BINDING PROTEIN-LIKE 5/7/9 (SPL5/7/9)*, all of which are floral activators [32–36], were downregulated following ethephon treatment (Table S5). Moreover, *TERMINAL FLOWER 1 (TFL1)*, a floral repressor, was upregulated following ethephon treatment [37] (Table S5).

In addition, TFs are vital proteins involved in crucial physiological functions in different tissues during different stages of development and physiological response. TFs can either repress or activate the expression of target genes, thereby modulating development and the physiological response [38,39]. In our analysis, most of the genes belonged to the following families of TFs: MYB, AP2-EREBP, bHLH, WRKY, and NAC (Figure S2). Many genes from the MYB and bHLH families are involved in floral transition [40]. In this study, we also found these two TF families were expressed differently after ethephon treatment (Figure 5b,c), which indicated that they may be involved in ethylene-mediated flowering regulation in chrysanthemum.

### 3.6. Validation of RNA-Seq Results Using qRT-PCR and Heterologous Expression in Arabidopsis thaliana

To verify the credibility of RNA-seq results, we selected four DEGs for qRT-PCR verification, including four ethylene signaling genes (*CTR1* and *EBF2*) and two flowering genes (*AFL1* and *TFL1*). Based on the results of qRT-PCR analysis, all four candidate DEGs showed expression patterns similar to those detected in RNA-seq (Figure 6a–d).

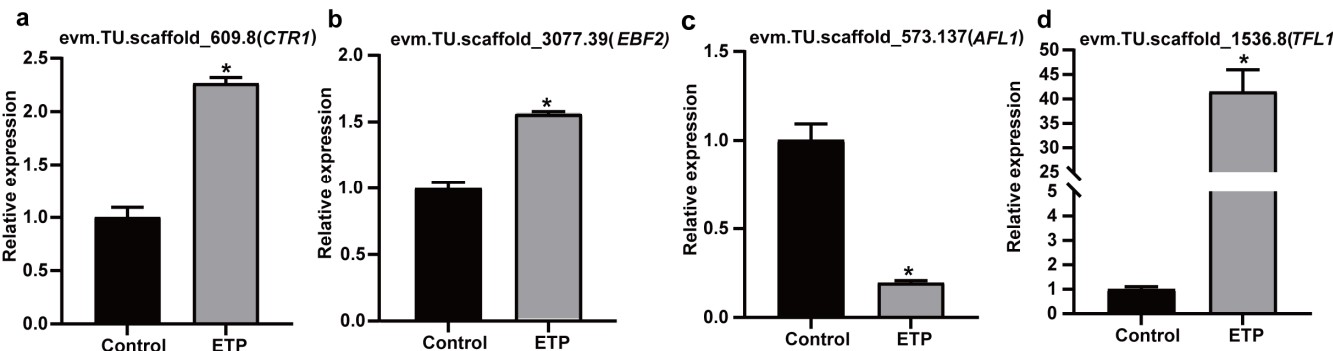

**Figure 6.** (**a**–**d**) qRT-PCR verification of differentially expressed genes (DEGs) associated with the ethylene signaling pathway and flowering regulation. *CmEF1α* was used as the loading control. The values are presented as mean ± standard deviation (*n* = 3). Transcript abundance was estimated using the $2^{-\Delta\Delta Ct}$ method. The significance of the differences was determined using Student's *t*-test (* $p < 0.05$).

*CmAFL1* is a known chrysanthemum flowering integrator [41]; however, no genetic evidence exists to support this conclusion. Therefore, we transformed the overexpression construct 35S::*CmAFL1* into wild type *A. thaliana (Col)* using the floral dip method [20]. Nine independent T$_1$ *A. thaliana* transgenic plants constitutively expressing *CmAFL1* were identified at the DNA level (Figure S3). When T$_3$ plants were obtained, we selected three overexpression lines (OX-1, OX-2, and OX-3) to observe the flowering phenotype using semi-quantitative RT-PCR assay (Figure 7a). Compared with *Col*, the transgenic plants showed earlier flowering, with fewer rosette leaves and shorter bolting time (Figure 7b–d). Therefore, *CmAFL1* promoted flowering in *A. thaliana*.

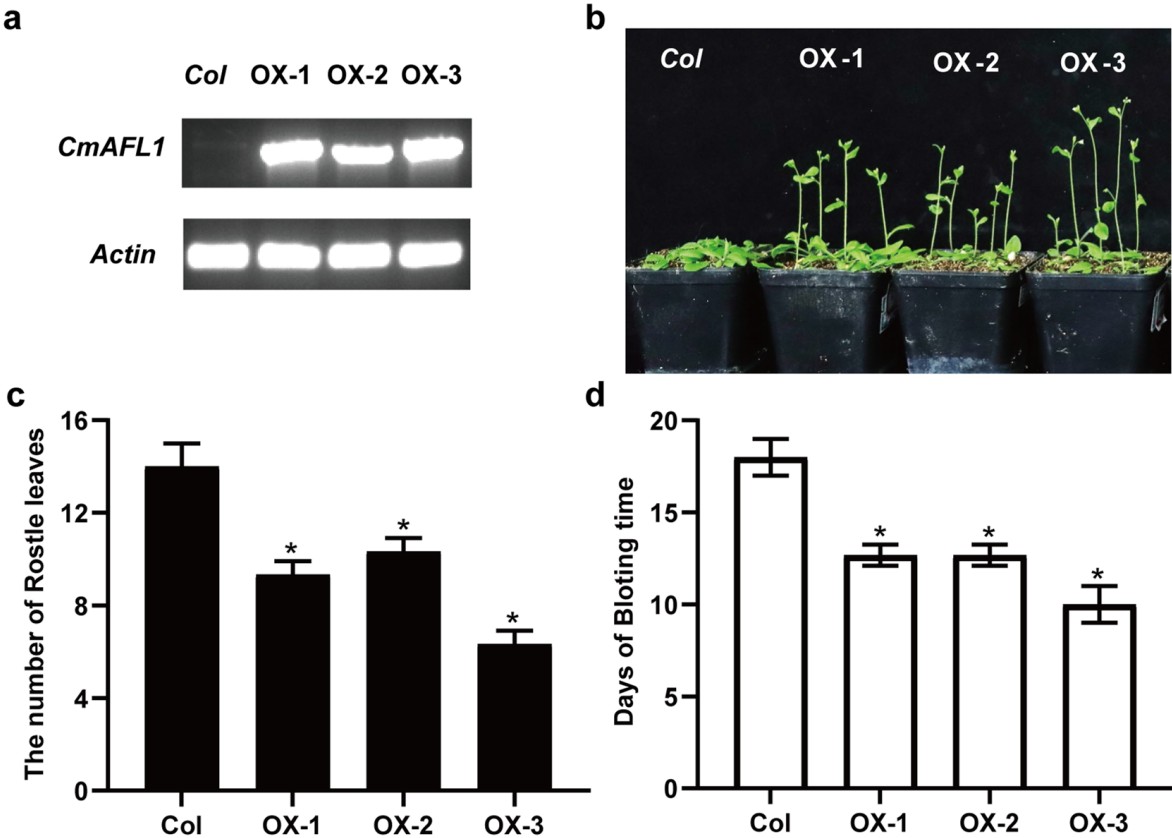

**Figure 7.** Phenotype following heterologous *CmAFL1* overexpression in *Arabidopsis thaliana*. (**a**) Results of semi-quantitative RT-PCR of *CmAFL1* in *Col* and *CmAFL1*-overexpressing plants. (**b**) Phenotype of 35S::*CmAFL1* plants at the bolting stage. Statistics of rosette leaves (**c**) and bolting time (days) (**d**) at the bolting stage. Values are presented as mean of ±standard deviation (*n* = 12). Significant differences were determined using Student's *t*-test (* *p* < 0.05). *Col*: wild type; OX-1 to OX-3: three independent overexpression lines.

## 4. Discussion

Ethylene-induced flowering has long been employed to facilitate year-round pineapple production, and the mechanism of ethylene-promoted flowering in pineapple has been revealed using RNA-seq [42,43]. Moreover, exogenous ethylene has been reported to suppress floral induction in chrysanthemum [44], although the molecular mechanism underlying ethylene-mediated flowering repression remains elusive. Therefore, in the present study, we constructed six transcriptome libraries of chrysanthemum 'Jinba' following water and ethylene treatments before floral induction and identified 2630 DEGs (Figure S1). These data will contribute to revealing the mechanism of ethylene-mediated flowering inhibition in chrysanthemum.

Through KEGG enrichment analysis, we identified some DEGs (*ETR2*, *CTR1*, and *EBF2*) involved in ethylene signaling, and each of them were upregulated after ethylene treatment (Figure 4a). Upregulation of *ETR2* and *EBF2* following ethylene treatment has also been described by Wang et al. [14] and Zhu et al. [45]. In rice, *ETR2* overexpression reduced ethylene sensitivity and delayed floral transition through upregulation of *GI* and a *TERMINAL FLOWER1/CENTRORADIALIS* (*RCN1*) homolog [16]. Similarly, in the present study, *ETR2* upregulation in chrysanthemum delayed flowering. *CTR1*, a Ser/Thr kinase (closely related to RAF kinases), negatively regulates ethylene signaling and works downstream of *ETR2*. The loss-of-function mutants of *CTR1* have been reported to exhibit late flowering in *A. thaliana* and the wild type *O. japonica* rice cultivar 'Dongjing' [17,18]. In contrast, in the present study, ethylene induced *CTR1* expression, while it delayed floral

transition. Therefore, the functions of *CTR1* may vary across plant species. Regarding *EBF2*, most recent studies have focused on its roles in plant senescence and fruit development [45–47], and there have been no studies on its possible involvement in flowering time regulation. Thus, whether *EBF2* regulates flowering via ethylene signaling warrants further exploration. ERF is a subfamily of the APETALA2/ethylene-responsive element-binding protein (AP2/ERF) superfamily, and its function in regulating flowering time has been rarely reported. In tobacco (*Nicotiana tabacum* L.), heterologous expression of tomato *SlERF36* led to early flowering [48]. *AtERF1* negatively modulated *FT* in *A. thaliana*, resulting in late flowering [49]. *CmERF110* collaborates with *FLOWERING LOCUS KH DOMAIN* (*CmFLK*) to accelerate chrysanthemum flowering [50]. Meanwhile, *CmERF3* inhibits the ability of *CmBBX8* to activate the flowering gene *CmFTL1* to inhibit flowering [51]. Overall, *ERF*s play diverse roles in the regulation of flowering time. In the present study, *ERF5* and *ERF113* were upregulated following ethylene treatment, whereas other ERFs, such as *ERF4* and *ERF8*, were downregulated following ethylene treatment (Figure 4a). Therefore, *ERF*s may play different roles in response to ethylene to regulate flowering. In addition, an enrichment of circadian clock genes was also found by KEGG analysis (Figure 3). Further analysis revealed that these circadian clock genes were upregulated in the ethephon-treated samples (Figure 5a), including the key rhythm clock output gene *GIGANTEA* (*GI*), which has been shown to be involved in photoperiodic flowering in chrysanthemum [52]. Therefore, we speculated that the expression of *GI* will be affected after ethephon treatment, which may interfere with the photoperiodic flowering of chrysanthemum.

In pineapple, a species with ethylene-promoted flowering, ethylene treatment upregulated the homologs of the auxin biosynthetic genes *WEAK ETHYLENE INSENSITIVE2 WEI 2* (*WEI 2*) and *WEAK ETHYLENE INSENSITIVE 8* (*WEI 8*) as well as of the ABA biosynthetic gene *VIVIPAROUS 14* (*VP14*), among others [14]. In contrast, DEGs related to auxin and ABA signaling were both downregulated after ethylene treatment in the present study (Figure 4b,c), consistent with the observed late-flowering phenotype after ethylene treatment in chrysanthemum. Overall, in chrysanthemum, ethylene may modulate auxin and ABA signaling to affect floral induction, further confirming the crosstalk among different phytohormones. These findings are consistent with previous reports. For instance, during leaf abscission, ethylene inhibits the synthesis and transportation of auxins or promotes their degradation, promoting the abscission process [53]. Under abiotic stress, ethylene reduces ABA accumulation, thereby alleviating heavy metal (As) stress in plants [54].

In differential gene expression analysis, we identified multiple genes whose homologs have been considered flowering genes, including genes that promote flowering, such as *AFL1* [55,56] and *SBPs* (*SBP1* and *SPL5/7/9*) [33,34,36,57], as well as genes that repress flowering, such as *TFL1* [37] (Table S5). *CmTFL1* suppressed flowering by directly downregulating *AtFT*, *AtLFY*, and *AtAP1* in *A. thaliana* [37]. Meanwhile, *CsTFL1* negatively regulated flowering by interfering with CsFTL3–CsFDL1 complexation in *Chrysanthemum seticuspe* f. *boreale* [58]. Furthermore, the co-expression of *CsTFL1* with CsFTL3/CsFDL1 complexation antagonists induced *CsAFL1* and *CsAFL2* expression [58]. Thus, whether ethylene promotes *CmTFL1* expression to increase *CmTFL1* levels or suppresses CmFTL3–CmFDL1 complexation to antagonize *CmAFL1*, ultimately delaying flowering, warrants further research. Although *CsAFL1* was recognized as a chrysanthemum flowering integrator, it was not genetically transformed into plants to confirm its function in floral transitions. In the present study, we noted that heterologous *CmAFL1* overexpression in *A. thaliana* could significantly accelerate flowering (Figure 7b–d), which is consistent with previous reports [55,56]. In the future, we will explore whether *CmAFL1* regulates chrysanthemum flowering via ethylene signaling.

## 5. Conclusions

In summary, the following preliminary conclusions are drawn. (I) Ethylene signaling genes may be involved in flowering regulation in chrysanthemum. (II) Ethylene may primarily affect auxin and ABA signaling to regulate floral induction. (III) Ethylene may

disrupt circadian rhythm to inhibit flowering. (IV) The floral activator *AFL1* may be the key gene affecting flowering following ethylene treatment. (V) Finally, MYB and bHLH TFs may be involved in ethylene-mediated flowering delay in chrysanthemum. Taken together, our work builds a foundation for further studies on the molecular machinery underlying ethylene-mediated flowering repression.

**Supplementary Materials:** The following supporting information can be downloaded at: https://www.mdpi.com/article/10.3390/horticulturae9040428/s1, Figure S1: Bar plot depicting the number of upregulated and downregulated differentially expressed genes between water and ethephon treatments; Figure S2. Number of unigenes from different transcription factor families; Figure S3. Identification of *CmAFL1* transgenic *Arabidopsis thaliana* at the DNA level. OX-1 to OX-9: nine independent transgenic lines. Col: wild type; $H_2O$: negative control. Primers: *CmAFL1*-F and GFP-R. Table S1: Primers used in the present study; Table S2. Statistics for sequencing; Table S3. Statistical comparison of sample sequencing data with the selected reference genomes; Table S4. All of the different expression genes. Table S5. Differentially expressed genes related to flowering time following ethephon treatment.

**Author Contributions:** Conceptualization, J.J. and H.C.; data curation, L.W.; validation, M.Z., Y.S. and W.L.; writing—original draft preparation, H.C.; writing—review and editing, S.C. and F.C.; funding acquisition, J.J. All authors have read and agreed to the published version of the manuscript.

**Funding:** This work was financially supported by grants from the National Natural Science Foundation of China (31930100), a project funded by the Priority Academic Program Development of Jiangsu Higher Education Institutions.

**Institutional Review Board Statement:** Not applicable.

**Informed Consent Statement:** Not applicable.

**Data Availability Statement:** All relevant data can be found in the manuscript and the Supplementary Materials. The RNA sequencing data from this study were uploaded to the NCBI SRA database (https://www.ncbi.nlm.nih.gov/bioproject/, accessed on 7 March 2023) with Bio Project ID: PRJNA939004. In this study, our reference genome is the hexaploid chrysanthemum cultivar 'Zhongshanzigui', which can be found at NCBI PRJNA796762.

**Conflicts of Interest:** The authors declare no competing interests.

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
