# Peer review of "Transcriptome Analysis of Ethylene Response in Chrysanthemum moriflolium Ramat. with an Emphasis on Flowering Delay"

_horticulturae, doi:10.3390/horticulturae9040428_

Round 1

Reviewer 1 Report

Authors should address some points:

1. Lines 29-34: Authors should add references in each sentence.

2. Lines 56-57: Authors should add references in this sentence.

3. Lines 71-82: Authors did not mention the chamber temperature in SD condition for chrysanthemum. Please check it. Authors use A. thaliana ecoltype as the wild type and grow them under long-day condition. Why did Authors not grow A. thaliana under SD condition as the same of chrysanthemum growth condition?

4. Line 84: Why did Authors select 100 mgL-1 ethephon for treatment? Did Authors try to test another concentration?

5. Line 90: "Three biological replicates were set per treatment". Why did you not select 9 replicates for each treatment?

Reviewer 2 Report

The paper from Cheng et al. presents a study of the transcriptome of Chrysanthemum moriflorum in response to ethylene, and the characterization of genes involved in flowering onset. In fact, unlike what the title suggests, this study extends far beyond flowering because it covers the wider range of ethylene response. This is why a better title would be: "Transcriptome analysis of ethylene response in Chrysanthemum moriflorum with an emphasis on flowering delay". 

The paper is well written and overall well-presented and there seem to be no concern with data. However, a major issue is that no extensive gene expression data is given. That could be useful to others. An Excel spreadsheet containing all 2630 DEF genes must be given as supplementary data, with gene annotation and expression parameters. Also, it would be interesting to know for each gene, the matching Chrysanthemum seticuspe gene ID, which can be found there: https://plantgarden.jp/en/list/t1111766. That would make this study really useful because people could retrieve any matching whole gene sequence from the C. seticuspe database. 

There are also a few minor issues that could be addressed:

- Line 76: Please specify what a "natural photoperiod" is? 24h30 should be 00h30.

- L. 78: define SD here, not line 84.

- L. 86-88: why are seedlings treated 5 times while samples are taken after the fourth treatment? If plants are treated every other day, the harvest after the 4th treatment should not be on day 6. Please clarify.

- L. 127: The construction of the 35S-CmAFL1 should be described: which 35S promoter? Which vector? How was the cloning made?

- Figure 1C: The x axis scale is misleading because not proportional. It should be replaced by a 0 to 120 continuous scale. 

- Fig. S1: This figure is useless as it is because it does not add any extra information over what is already given in the text. It should be replaced by a graph showing the number of down or upregulated genes at fold change x1.5, x2, x3, x4, x5... x10 or more.

Round 2

Reviewer 2 Report

The authors have addressed most requests. However, their description of the cloning procedure to make the 35S-CmAFL1 construct is not correct. The authors mention the pORE-R4 vector without giving any reference for it. A reference for the plasmid can be found here:

https://www.ncbi.nlm.nih.gov/nucleotide/AY562547

It appears that this plasmid is not an overexpression vector but a promoter-analysis vector. It contains the gfp coding sequence and some polylinkers but no 35S promoter. Therefore, the method given by the authors cannot be correct. Where did the 35S promoter which is said to be in their final construct come from? The correct method for the construction of the plasmid bearing the 35S-CmAFL1 fusion must be given. 
